# NGF-Enhanced Vasculogenic Properties of Epithelial Ovarian Cancer Cells Is Reduced by Inhibition of the COX-2/PGE_2_ Signaling Axis

**DOI:** 10.3390/cancers11121970

**Published:** 2019-12-07

**Authors:** Maritza P. Garrido, Iván Hurtado, Manuel Valenzuela-Valderrama, Renato Salvatierra, Andrea Hernández, Margarita Vega, Alberto Selman, Andrew F. G. Quest, Carmen Romero

**Affiliations:** 1Laboratorio de Endocrinología y Biología de la Reproducción, Hospital Clínico Universidad de Chile, Santiago 8380456, Chile; mgarrido@hcuch.cl (M.P.G.); ihurtado@hcuch.cl (I.H.); rsalvatierratm@gmail.com (R.S.); ahernandezj@hcuch.cl (A.H.); mvega@med.uchile.cl (M.V.); 2Departamento de Obstetricia y Ginecología, Facultad de Medicina, Universidad de Chile, Santiago 8380453, Chile; selmanalberto@gmail.com; 3Laboratorio de Microbiología Celular. Instituto de Investigación e Innovación en Salud. Facultad de Ciencias de la Salud, Universidad Central de Chile, Santiago 8320000, Chile; manuel.valenzuela@ucentral.cl; 4Advanced Center for Chronic Diseases (ACCDIS), Santiago 8380000, Chile; 5Laboratorio de Comunicaciones Celulares, Centro de estudios en Ejercicio, Metabolismo y Cáncer (CEMC), Facultad de Medicina, Universidad de Chile, Santiago 8380453, Chile

**Keywords:** NGF, epithelial ovarian cancer, COX-2/PGE_2_, vasculogenesis, VEGF, c-MYC, survivin, beta-catenin

## Abstract

Epithelial ovarian cancer (EOC) is a lethal gynecological neoplasia characterized by extensive angiogenesis and overexpression of nerve growth factor (NGF). Here, we investigated the mechanism by which NGF increases vascular endothelial growth factor (VEGF) expression and the vasculogenic potential of EOC cells, as well as the contribution of the cyclooxygenase 2/prostaglandin E_2_ (COX-2/PGE_2_) signaling axis to these events. EOC biopsies and ovarian cell lines were used to determine COX-2 and PGE_2_ levels, as well as those of the potentially pro-angiogenic proteins c-MYC (a member of the Myc transcription factors family), survivin, and β-catenin. We observed that COX-2 and survivin protein levels increased during EOC progression. In the EOC cell lines, NGF increased the COX-2 and PGE_2_ levels. In addition, NGF increased survivin, c-MYC, and VEGF protein levels, as well as the transcriptional activity of c-MYC and β-catenin/T-cell factor/lymphoid enhancer-binding factor (TCF-Lef) in a Tropomyosin receptor kinase A (TRKA)-dependent manner. Also, COX-2 inhibition prevented the NGF-induced increases in these proteins and reduced the angiogenic score of endothelial cells stimulated with conditioned media from EOC cells. In summary, we show here that the pro-angiogenic effect of NGF in EOC depends on the COX-2/PGE2 signaling axis. Thus, inhibition COX-2/PGE2 signaling will likely be beneficial in the treatment of EOC.

## 1. Introduction

Epithelial ovarian cancer (EOC) is the eighth largest cause of death by cancer in women worldwide [1,2]. This pathology is characterized by non-specific symptoms and, therefore, is diagnosed at later stages, resulting in poor survival rates [3,4,5,6]. One important characteristic of this neoplasm is the high extent of angiogenesis that facilitates rapid tumor growth and dissemination [7]. The most studied angiogenic factor, vascular endothelial growth factor (VEGF), is overexpressed in EOC [7,8]. Our group has shown that nerve growth factor (NGF) and its high affinity receptor tropomyosin receptor kinase A (TRKA) increase during EOC progression [9], activating phosphoinositide 3-kinase/ protein kinase B (PI3K/AKT) and mitogen-activated protein kinase /extracellular signal-regulated kinase (MAPK/ERK) signaling pathways, resulting in enhanced tumor growth [10]. In addition, NGF represents both a direct and indirect angiogenic factor [9,11,12], as NGF increases VEGF levels in EOC cells [9,11] and also binds to the high affinity receptor TRKA in endothelial cells, thereby increasing their proliferation, migration, and vasculogenesis [9,12].

To shed light on the mechanism(s) by which NGF increases VEGF expression in EOC cells, we studied some known regulators of VEGF expression, which include cyclooxygenase 2/prostaglandin E_2_ (COX-2/PGE_2_) [13,14,15,16], survivin (BIRC5) [17,18], the β-catenin/ T-cell factor/lymphoid enhancer-binding factor (TCF-Lef) transcription complex [17,19,20], and the c-MYC transcription factor [21,22]. COX-2 is a key enzyme in the synthesis of prostaglandins, including PGE_2_, the most commonly and ubiquitously produced derivative [23,24]. COX-2 can be induced by various stimuli and plays a key role in inflammation [23,24,25] and cancer progression [26,27,28,29]. Different in vitro studies have demonstrated that COX-2 and PGE_2_ induce VEGF secretion in an ERK-dependent manner [15,30,31,32]. On the other hand, EOC is characterized by overexpression of oncogenic proteins, such as c-MYC and β-catenin, which are associated with poor prognosis [33,34,35] and are known to increase the transcription of VEGF and survivin [36,37,38]. Interestingly, a positive loop between β-catenin/TCF-Lef, COX-2, and VEGF has been described in melanoma and gastric cells involving the PI3K/AKT signaling pathway [17,39]. All these regulatory mechanisms are likely to cooperate in promoting angiogenesis in EOC.

Previous findings indicate that NGF/TRKA increase c-MYC and VEGF expression in EOC explants [10]. Therefore, we evaluated whether NGF stimulation enhances COX-2 protein levels and PGE_2_ in EOC cells, and also, the expression of the pro-angiogenic proteins c-MYC, β-catenin, survivin, and VEGF. In addition, we evaluated the dependence of NGF-effects on signaling via the COX-2/PGE_2_ axis.

Our results show that COX-2 and survivin protein levels increase during EOC progression, being higher in advanced stages of the disease. NGF stimulation of EOC cells (A2780, SKOV3, OVCAR3, and OV90) increases COX-2 and PGE_2_ levels, the expression of the pro-angiogenic proteins survivin, c-MYC, and VEGF, as well as the transcriptional activity of c-MYC and β-catenin/TCF-Lef. Direct stimulation of EOC cells with PGE_2_ yielded comparable results to those observed with NGF. Moreover, in the presence of a COX-2 inhibitor, NGF-induced effects were prevented. These results suggest that the NGF-induced expression of pro-angiogenic proteins and the angiogenic potential of EOC cells depend on signaling via the COX-2/PGE_2_ axis.

## 2. Results

### 2.1. COX-2 Expression Increases During EOC Progression

Samples of well-differentiated epithelial ovarian cancer (EOC I), moderately differentiated epithelial ovarian cancer (EOC II), and poorly differentiated epithelial ovarian cancers (EOC III) were grouped as EOC, whereas benign and borderline tumors were grouped as ovarian tumors (OvTu). Of note, COX-2 protein and its messenger RNA (mRNA) levels were significantly higher in the EOC group as compared to the inactive ovarian epithelium (IOV) group (*p* < 0.05; Figure 1A–D). Likewise, COX-2 protein levels were higher in the EOC group compared with the IOV group (*p* < 0.05; Figure 1C). Immunohistochemical analysis identified COX-2 in epithelial cell monolayers and transformed epithelial cells, whereby staining was mainly cytoplasmic (Figure 1E). Additionally, during EOC progression, a substantial increase in COX-2 levels was observed, and this increase became significant at the borderline tumor stage (BorT) (*p* < 0.01 vs. IOV; Figure 1E).

### 2.2. NGF Increases COX-2 Expression in EOC Cells

Basal COX-2 levels in EOC cell lines (A2780, SKOV3, OV90, and OVCAR3 cell lines) were higher than in non-tumoral human ovarian surface epithelium (HOSE) cells (Figure 1F). Upon stimulating with NGF (100 and 150 ng/mL), significant increases in COX-2 protein levels were observed in EOC cells, but not in non-tumoral HOSE cells (*p* < 0.05 and *p* < 0.01; Figure 1G and Appendix A). Because transforming growth factor beta (TGF-β) has been shown to induce PGE_2_ in granulose cells [40], it was used as a positive control. A significant increase in COX-2 mRNA and protein levels was detected in A2780 cells stimulated with 15 or 20 ng/mL of TGF-β1 (*p* < 0.05; Appendix A). Furthermore, the increase in COX-2 protein levels in A2780 cells triggered by TGF-β1 correlated with accumulation of PGE_2_ in culture supernatants (*p* < 0.05; Appendix A).

### 2.3. NGF Increases PGE_2_ Secretion by EOC Cells

To determine whether the COX-2 increase observed in EOC cells lead to enhanced secretion of prostaglandin E_2_, levels of these prostanoids were determined in the conditioned culture media of ovarian cells. NGF stimulation induced an increase in the presence of such prostaglandins in the cell culture medium of all EOC cell lines (*p* < 0.05 in A2780 and OVCAR3 cells, *p* < 0.01 in SKOV3 and OV90 cells; Figure 1H). The mean increase was higher in A2780 cell lines, which may be because baseline levels of PGE_2_ in these cells are lower compared with the other EOC cell lines (Appendix A). Alternatively, the increase in PGE_2_ induced by NGF was observed only in EOC cells. NGF treatment of non-tumoral HOSE cells did not increase significantly PGE_2_ levels in the conditioned medium of these cells (Appendix A).

### 2.4. Role of COX/2-PGE_2_ in VEGF Secretion by Ovarian Cells

Previous results demonstrated that NGF increases VEGF expression in both EOC tissue explants and A2780 cells [9,11]. Thus, considering that in other cancers PGE_2_ is known to modulate VEGF levels [13,14,15,16], we evaluated here whether such NGF-mediated increases in VEGF were connected to the COX-2/PGE_2_ signaling pathway. As Figure 1I shows, NGF treatment (100 or 150 ng/mL) increased VEGF protein levels in all EOC cell lines (*p* < 0.05, *p* < 0.01, and *p* < 0.001 for SKOV3, OV90/OVCAR3, and A2780, respectively). Importantly, the increase in VEGF triggered by NGF was reduced significantly by prior treatment of cells with a specific COX-2 inhibitor (NS, 20 µM) (*p* < 0.05 in OV90 and OVCAR3 cells; and *p* < 0.01 in A2780). These results are consistent with the notion that COX-2 inhibition blocks the NGF-induced increase in PGE_2_ in the EOC cell line A2780 (Appendix A), which suggests that NGF increases VEGF levels in a COX2/PGE_2_-dependent manner in EOC cells.

Because many proteins and transcription factors regulate VEGF expression [17,18], we sought to identify possible mechanisms by which NGF-COX-2/PGE_2_ signaling increased VEGF levels in ovarian cells.

### 2.5. Survivin Levels Increase During EOC Progression

Survivin is an oncogenic protein that is associated with poor prognosis in ovarian cancer [41] and reportedly increases VEGF levels in in vitro models [17,18]. Indeed, for EOC and OvTu samples, higher levels of survivin were observed by immunohistochemistry (IHQ) compared to IOV samples (*p* < 0.001 and *p* < 0.0001, respectively; Figure 2A,B). Similar results were obtained when survivin protein levels in EOC samples were evaluated by western blotting (*p* < 0.01; Figure 2C,D). In both cases, increases were associated with EOC progression (Appendix A). In addition, basal survivin levels assessed in ovarian cell lines (Figure 2E), revealed that EOC cell lines had higher levels of survivin compared with non-tumoral HOSE cells. These findings are consistent with the biopsy results, confirming that these cell lines represent appropriate models to study survivin in the context of EOC.

### 2.6. NGF Increases Survivin Levels in Epithelial Ovarian Cells

NGF stimulation of ovarian cell lines significantly increased survivin protein levels compared with the baseline condition (*p* < 0.05 in HOSE and A2780 cells; *p* < 0.01 in SKOV3, OV90, and OVCAR3; Figure 2F–K). Importantly, this increase was blocked by the specific TRKA inhibitor GW441756 or an NGF neutralizing antibody (Figure 2F–H), indicating that the NGF- mediated effect was mediated by the high-affinity NGF receptor TRKA.

### 2.7. NGF Increases β-Catenin-TCF-Lef Transcriptional Activity in EOC Cells

Association of β-catenin with TCF-Lef transcription family members in the nucleus generates a complex that increases the transcription of many oncogenic proteins and factors, including VEGF [17,19,20]. Basal levels of β-catenin did not differ substantially between the EOC cell lines (Figure 3A) and NGF stimulation of EOC cells induced a modest increase in β-catenin protein levels in SKOV3 and OV90 cells (Figure 3B and Appendix A). However, following NGF stimulation, higher cytoplasmic and/or nuclear levels of β-catenin were detected (Figure 3C). Then β-catenin/TCF-Lef reporter assays were performed in EOC cell lines. We observed that NGF enhanced β-catenin/TCF-Lef transcriptional activity in the four EOC cell lines studied (*p* < 0.05; Figure 3D). Furthermore, the NGF-dependent increase in β-catenin/TCF-Lef transcriptional activity in A2780 cells was blocked by the specific TRKA inhibitor GW441756 (*p* < 0.05; Appendix A). Taken together, these results show that NGF/TRKA signaling augments the transcriptional activity of β-catenin/TCF-Lef in EOC cells.

### 2.8. NGF Increases c-MYC Levels and Its Transcriptional Activity in EOC Cells

c-MYC is an important proto-oncogene and previous reports have shown that c-MYC activation can increase VEGF expression [21,22]. As determined by western blotting shown in Figure 4A, basal c-MYC levels were notably different in ovarian cell lines, being elevated in A2780 and OVCAR3 cells and essentially undetectable in HOSE cells. In addition, the results showed that NGF stimulation increased c-MYC protein levels in all ovarian cell lines used (*p* < 0.05 in SKOV3 and OVCAR3 cells, *p* < 0.01 in A2780 and OV90 cells, and *p* < 0.001 in HOSE cells; Figure 4B–D). Importantly, the NGF-mediated increase in c-MYC levels was prevented either by addition of the TRKA inhibitor (GW441756) or an NGF neutralizing antibody (Figure 4B,C). Moreover, NGF stimulation induced nuclear accumulation of c-MYC in EOC cells (*p* < 0.05 in SOV3 and OVCAR3 cells; *p* < 0.01 in A2780 and OV90 cells; Figure 4E), as well as increased c-MYC transcriptional activity (*p* < 0.05; Figure 4F) in a TRKA-dependent manner (Appendix A). These results confirm in cell lines the previous reports from our group, showing that NGF stimulates c-MYC expression in EOC explants [10].

### 2.9. Inhibition of the COX-2/PGE_2_ Axis Prevents NGF Stimulated Increases in Pro-Angiogenic Proteins in EOC Cells

To determine whether NGF stimulation increases the expression of pro-angiogenic proteins through a COX-2/PGE_2_-dependent mechanism, EOC cells were pre-incubated with the specific COX-2 inhibitor NS398 (20 μM) at a concentration that blocks the NGF-stimulated PGE_2_ increases (Appendix A), and then EOC cells were stimulated with NGF (100 or 150 ng/mL), as was indicated in the Methodology section. The results showed that inhibition of COX-2 prevented the NGF-induced increase in survivin and c-MYC protein levels in all EOC cell lines (*p* < 0.05 Figure 5A–D), as well as the NGF-induced VEGF expression (see Figure 1I). The COX-2 inhibitor did not affect baseline β-catenin protein levels (Figure 5A–D and Appendix A); however, as expected, the inhibition of COX-2 prevented the NGF-induced β-catenin/TCF-Lef and MYC transcriptional activity in most cell lines (*p* < 0.05; Figure 5E–H).

To confirm that COX-2 modulates the angiogenic potential of EOC cells, we performed an in vitro assay with endothelial cells. Conditioned media from EOC cells stimulated with NGF increased the angiogenic score of endothelial cells (*p* < 0.05; Figure 5I–L). Alternatively, EOC cells pre-treated with COX-2 inhibitor and stimulated with NGF were unable to increase the angiogenic score of EA.hy926 cells compared with the NGF-stimulated group (*p* < 0.05 for OVCAR3 cells, *p* < 0.01 for SKOV3 cells, and *p* < 0.001 in A2780 cells). Also, for culture supernatants from non-tumoral HOSE cells, lower angiogenic scores were observed compared with EOC cells and, moreover, COX-2 inhibition of HOSE cells blocked the NGF-induced increase in angiogenic score (Appendix A).

Taken together, these results demonstrate that the increase in VEGF and the vasculogenic potential of EOC cells stimulated by NGF depends on activation of the COX-2/PGE_2_ signaling axis.

### 2.10. PGE_2_ Stimulation Increases the Expression of Pro-Angiogenic Proteins in EOC Cells

Because COX-2 inhibition prevents the NGF-mediated increases of all pro-angiogenic proteins evaluated, the effects of stimulating cells with PGE_2_ were also determined. As shown in Figure 6, PGE_2_ stimulation increased c-MYC and survivin protein levels (*p* < 0.05 and *p* < 0.01; Figure 6A–D). In addition, PGE_2_ increased β-catenin protein levels in only two of four EOC cell lines (Appendix A). However, PGE_2_ increased β-catenin/TCF-Lef transcriptional activity in all EOC cell lines (*p* < 0.05; Figure 6E–H), as well as VEGF levels in the culture supernatants of these EOC cells (*p* < 0.05 and *p* < 0.01; Figure 6I–L). Finally, as expected, PGE_2_ increased MYC transcriptional activity in EOC cells (*p* < 0.05, Appendix A).

## 3. Discussion

EOC is the leading cause of death due to gynecological neoplasms in developed countries [5]. The high mortality and the poor response to therapies in this pathology [5,7] require a better understanding of the molecular mechanisms implicated in the development of EOC in order to identify new therapeutic targets. Previous results have shown that NGF/TRKA are involved in the progression and angiogenesis in EOC [9]. The current findings reveal that NGF/TRKA enhances expression of the pro-inflammatory enzyme COX-2 and PGE_2_ secretion in EOC cells. Importantly, we show that COX-2/PGE_2_ signaling regulates the levels of several pro-angiogenic proteins, such as c-MYC, survivin, the β-catenin/TCF-Lef transcription complex, and VEGF. This suggests that the pro-inflammatory COX-2/PGE_2_ axis plays a key role in the progression and dissemination of EOC and, therefore, represents an attractive therapeutic target in this pathology.

NGF is an important pro-tumoral factor, because it not only increases the proliferation of EOC cells [10,12] but also is proposed to function as a direct and indirect angiogenic factor in EOC, given that it increases VEGF secretion in EOC explants [9,11] and can bind to the TRKA receptor in the endothelial cells to promote angiogenesis [9,12]. Our results reinforce these previous findings, as stimulation of EOC cell lines with NGF increases VEGF secretion and increases their vasculogenic potential. In the present study, we sought to shed light on the cellular mechanism by which NGF increases VEGF secretion. For this reason, we evaluated changes in proteins and transcription factors implicated in increasing VEGF expression in several models of cancer cells, such as c-MYC, β-catenin, and survivin. Previous reports from our group have shown that NGF increases c-MYC levels in EOC explants [10]; however, the effect of NGF on the other angiogenic proteins mentioned above were unknown. Our results show that NGF increases c-MYC and survivin levels in non-tumoral HOSE and EOC cell lines, as well as the transcriptional activity of MYC and β-catenin/TCF-Lef in EOC cells. These observations suggest that NGF not only has a pro-angiogenic effect in EOC cells, but may also contribute to the transformation of non-tumoral ovarian epithelial cells.

One interesting point here is that NGF-stimulation increased survivin levels in EOC cells, which has not been described previously. Survivin is an important protein in cancer that favors the survival of tumor cells by inhibiting cell death and promoting cell proliferation [42]. The results obtained in HOSE cells revealed a significant increase in survivin after short stimulation (2 h post-NGF stimulation), whereas in EOC cells the greatest increases in survivin levels were observed after longer time periods (8 and 24 h after NGF stimulation). This difference could be explained by a variety of mechanisms. Survivin is a short-lived protein with a half-life of about 30 min, and proteasome inhibitors greatly stabilize survivin in vivo [43]. In addition, survivin levels increase following EGF stimulation of the ERK-depending pathway for short periods (2 and 4 h) in beta pancreatic cells. These observations suggest that the rapid increase in survivin could be mediated by NGF-dependent inhibition of proteasome-mediated degradation in ovarian cells. On the other hand, other authors have described that ERK and AKT signaling is required to activate survivin transcription in colorectal cancer cells [44]. Thus, the long-term increase in survivin levels observed in EOC cells mediated by NGF/TRKA could potentially be attributed to an increase in survivin transcription, which was evidenced by an increase in survivin mRNA post-NGF stimulation in ovarian cells (Appendix A).

Because COX-2/PGE_2_ signaling has been linked to increases in VEGF in many in vitro models, we determined whether NGF-enhanced expression of pro-angiogenic proteins occurs through activation of the COX-2/PGE_2_ system. The overexpression of COX-2 has been shown to be highly relevant in various neoplasms, such as colon and stomach cancers [26,45]. In the present study, as has been described by others [46,47], significantly increased COX-2 immunostaining was observed in EOC tissue, compared with samples from early stages of ovarian cancer. The same pattern was observed at the protein and mRNA levels for COX-2. These results point towards the existence of a correlation between COX-2 overexpression and several clinical variables, such as size, histological type, and tumor grade [48,49]. COX-2 overexpression correlates with drug resistance and reduced patient survival [48,49], whereas NGF/TRKA has been implicated in EOC progression [9]. Thus, our results suggest that resistance to chemotherapy and low survival rates of advanced EOC patients might be attributable to elevated levels of COX2 and that use of specific inhibitors could be helpful as part of the EOC therapy.

NGF-induced increases in COX-2 protein and mRNA levels, as well as PGE_2_ secretion, were observed in EOC cells. In order to determine whether NGF effects on angiogenic proteins (c-MYC, survivin, and VEGF) depended on COX-2/PGE_2_ signaling, we evaluated the effects of a COX-2-specific inhibitor and direct stimulation with PGE_2_. Our results show that pre-treatment of EOC cells with the COX-2 inhibitor precluded augmented expression of all pro-angiogenic proteins induced by NGF. On the other hand, EOC cells stimulated with PGE_2_ showed an increase in c-MYC, survivin, and VEGF protein levels, and increased β-catenin/TCF-Lef-dependent transcriptional activity was also detected. Together, these results show that the pro-angiogenic effects of NGF strongly depend on signaling via the COX-2/PGE_2_ axis in EOC cells. Somewhat surprisingly, however, NGF-increased VEGF levels were not prevented by pre-treatment with the COX-2 inhibitor for short periods of time (Appendix A). Another crucial point is that NGF increases VEGF in non-tumoral HOSE cells independently of COX-2/PGE_2_ signaling, as NGF did not increase COX-2 or PGE_2_ in this cell type. These results suggest that NGF produces an increase in VEGF in ovarian cells by at least two different mechanisms: (1) a rapid increase, independent of the COX-2/PGE_2_ axis, perhaps mediated by an increase in VEGF release [50] or a rapid increase in VEGF mRNA, as described in EOC explants [11]; (2) a slower response mediated by COX-2/PGE_2_, which appears to be crucial to increased VEGF expression in EOC cells. This interpretation of the results is favored by the fact that PGE_2_ needs to be synthesized and released from cells, and then to bind to cell surface membrane receptors, known to be present in ovarian cells [51].

Interestingly, inhibition of endogenous expression of angiogenic proteins was not observed using TRKA or COX-2 inhibitors. For instance, survivin is regulated by several growth factors and their respective receptors, including epidermal growth factor (EGF) and fibroblast growth factor (FGF) in breast cancer cells [36,52]. These growth factors/receptors are present in ovarian cancer cells, so it is possible that survivin could be regulated by growth factors other than NGF, especially in a compensatory manner when NGF is inhibited. Another interesting observation is that COX-1 is aberrantly expressed in ovarian tumors and ovarian cancer, especially in high grade serous ovarian cancer cells [53,54]. Thus, under conditions where COX-2 is inhibited, it is possible that levels of pro-angiogenic proteins are maintained thanks to the contribution of COX-1.

To determine whether the inhibition of COX-2/PGE_2_ signaling and decreases in VEGF secretion leads to significant changes in the angiogenic potential of EOC cells, we evaluated the behavior of endothelial cells in vasculogenesis assays. Angiogenesis is a complex process that involves different cell types, the extracellular matrix, and humoral components such as growth factors [55]. However, in vitro studies evaluating vasculogenesis induced by different stimuli in endothelial cells, serve to provide insight to the angiogenic potential of cancer cells [56]. The NGF-induced increase in the angiogenic score of EA.hy926 cells was prevented by pre-incubation of EOC cells with the COX-2 inhibitor. This functional assay reinforces our key finding indicating that increases in VEGF due to NGF in EOC are triggered via the COX-2/PGE_2_ axis.

In summary (see Figure 7), the present work describes for the first time a connection between NGF and COX-2/PGE_2_ signaling in EOC cells. NGF was found to increase levels of the oncogenic proteins survivin, c-MYC, and VEGF, as well as the transcriptional activity of β-catenin/TCF-Lef and MYC in EOC cells. Moreover, these effects were mediated by the COX-2/PGE_2_ axis, highlighting the importance of this connection between inflammation and angiogenesis in EOC. Taken together, our results suggest that inhibition of the COX-2/PGE_2_ axis could be helpful to complement the currently employed anti-angiogenic therapies in EOC.

## 4. Methods

### 4.1. Tissue Samples

Ovarian tissue samples were removed surgically from post-menopausal patients at the Hospital Clínico Universidad de Chile. Each patient signed an informed consent approved by the Institutional Ethics Committee (Comité de Ética Hospital Clínico Universidad de Chile, in 18 May 2016), Record N°022, 2016). Sequential Paraffin-embedded samples were classified as inactive normal ovarian tissue samples (IOV, *n* = 4), serous ovarian tumors (benign tumor, Be-T, and borderline tumor, Bo-T, *n* = 15), or serous epithelial ovarian cancer (EOC *n* = 10), and were used for immunohistochemistry analysis. EOC samples were classified as well (EOC I), moderately (EOC II), or poorly (EOC III) differentiated by an expert pathologist.

### 4.2. Cell Culture and Treatments

Human ovarian surface epithelium (HOSE) cells were donated by Dr. Davie Munroe (NCI, NHI) and human epithelial ovarian cancer cell lines A2780 were obtained from the European Collection of Authenticated Cell Cultures (ECACC). SKOV3, OV90, and NIH-OVCAR3 (OVCAR3) cells were obtained from the American Type Culture Collection (ATCC). A2780 and HOSE cell lines were cultured in Dulbecco’s minimal essential medium/Ham F-12 without phenol red (Sigma-Aldrich, St. Louis, MO, USA), whereas SKOV3, OV90, and OVCAR3 cells were cultivated in Roswell Park Memorial Institute (RPMI)-1640 medium (Life Technologies, Thermo Fisher Scientific, Carlsbad, CA, USA). Both culture media were supplemented with 10% fetal bovine serum (FBS) and penicillin/streptomycin (100 µg/mL and 100 U/mL). The human endothelial cell line EA.hy926 was obtained from ATCC Cell Collection and was cultured in Iscove’s Modified Dulbecco’s Medium (IMDM) (Life Technologies, Thermo Fischer Scientific) supplemented with 10% FBS. A total of 500,000 cells from both ovarian cell lines were serum-deprived for 24 h and treated with NGF (Sigma-Aldrich) as indicated: (1) dose-response curves with 50, 100, and 150 ng/mL of NGF in the first experiments and (2) NGF at a concentration of 100 ng/mL (HOSE, A2780, and OV90 cells) or 150 ng/mL (SKOV3 and OVCAR3 cells) for the remaining experiments. In addition, EOC cells were treated with a TRKA inhibitor GW441756 20 nM (Tocris, Bristol, UK), an NGF-neutralizing antibody (Abcam #ab6199, 5 µg/mL, Cambridge, UK), the specific COX-2 inhibitor NS398 (Abcam, 20 µM), or prostaglandin E2 (Sigma, 20 µM) for 2 or 24 h. Cells were pre-incubated with the inhibitors for 1 h prior to addition of NGF.

### 4.3. Immunohistochemistry (IHQ)

IHQ was performed as previously described [11]. Anti-COX-2 (Cell signaling Technology #12282; 1:600, Danvers, CO, USA) or anti-survivin antibodies (R&D Systems #AF886; 1:500, Minneapolis, MS, USA) were incubated at 4 °C overnight. For each slide, a negative control incubated only with 2% phosphate buffered saline-bovine serum albumin (PBS-BSA) without the primary antibody was included. Five to ten microphotographs were obtained with each sample. Immunostaining was semi-quantified by integrated optical density (IOD) analysis of positive cells using the program Image-ProPlus 6.2 (Media Cybernetics, Rockville, MD, USA).

### 4.4. Immunocytochemistry (ICQ)

ICQ was performed as previously described [12]. c-MYC (Cell Signaling #5605, 1:500) and β-catenin (BD Transduction Laboratories #610154, 1:1000, Franklin Lakes, NJ, USA) were detected and IOD values were obtained as described above.

### 4.5. Western Blotting

Nitrocellulose membranes with protein extracts (30 or 50 μg of total protein) were incubated with either anti-COX-2 (Cell Signaling #12282, 1:500) or anti-survivin antibodies (R&D Systems #AF886) at dilutions of 1:3000 for A2780, SKOV3, OV90, and OVCAR3 cells, and 1:1000 for HOSE cells (because of lower expression in this cell type), anti-c-MYC (#5605, 1:500), or anti-β-catenin antibodies (BD Transduction Laboratories #610154; 1:3000) at 4 °C overnight. Beta-actin (Sigma #A2228, 1:10,000) was used as loading control. Human embryonic kidney (HEK)-293 cells transfected with COX-2 or survivin, were used as positive controls. Protein bands were quantified by scanning densitometry of western blots using the software Fiji Image J (developed by National Institutes of Health/University of Wisconsin, WI, USA). Original captures of western blots were included in Appendix A.

### 4.6. Total RNA Extraction and RT-PCR

Total RNA extraction was performed using the phenol-chloroform method. A total of 2 µg of RNA was used for reverse transcription, as described previously [57]. COX-2 and glyceraldehyde 3-phosphate dehydrogenase (GAPDH) cDNA amplification were performed by traditional PCR using GoTaq polymerase (Promega, Madison, WI, USA). Survivin amplification was performed by real time PCR using the Brilliant II SybrGreen QPCR master mix (Agilent Technologies, Santa Clara, CA, USA). All primers used are described in Appendix A. GAPDH or beta-actin were used as invariant control genes. Sterile water instead of cDNA was added as a negative control for the reactions.

### 4.7. PGE_2_ and VEGF ELISA

PGE_2_ and VEGF in culture supernatants were determined with ELISA kits (Abcam #Ab136948 and R&D Systems #DVE00 respectively), according to the manufacturer’s instructions.

### 4.8. β-catenin/Tcf-Lef Reporter Assay

Cells (450,000) were transfected for 8 h using 0.3 μL of ViaFect (Promega) in 50 μL of opti-MEM^®^ I reduced serum media (Thermo Fisher Scientific) and 0.5 μg of the following plasmids: (1) pTOP-FLASH (that contains β-catenin/Tcf-Lef responsive elements), (2) pFOP-FLASH (non-inducible reporter construct), and (3) pPON-FLASH (beta-galactosidase transfection control). Then, cells were deprived of FBS overnight and stimulated the following morning as described in Section 4.2. Cells were lysed and luciferase activity was measured with the Dual luciferase reporter assay system (Promega) according to the manufacturer’s instructions. LiCl 15 mM was used as a positive control. β-galactosidase activity of constructs was measured as previously described [58].

### 4.9. MYC Reporter Assay

A total of 30,000 cells in 24-well plates were used. The experiment was performed using the Cignal Myc Reporter Assay Kit (Qiagen, Hilden, Germany), according the manufacturer’s instructions.

### 4.10. Vasculogenesis Assay

A total of 10,000 EA.hy926 cells were serum-deprived for 24 h and then placed in 24-well plates that were coated with 150 µL of growth factor reduced, phenol red-free Matrigel (Corning, New York, NY, USA) and 500 µL of the different conditioned media from A2780 cells. After 8 h, cells were photographed and the angiogenic score was calculated [59]. This index is based on the morphology of and the connections between endothelial cells and was calculated using the following formula:$$\mathrm{Angiogenic}\;\mathrm{Score}=\frac{N{}^{\circ}\;\mathrm{of}\;\mathrm{sprouts}+\left(N{}^{\circ}\;\mathrm{of}\;\mathrm{connected}\;\mathrm{cells}\right)\times 2+\left(N{}^{\circ}\;\mathrm{of}\;\mathrm{polygons}\right)\times 3}{N{}^{\circ}\;\mathrm{of}\;\mathrm{total}\;\mathrm{cells}}+0,\;1,\;\mathrm{or}\;2.$$

### 4.11. Statistical Analysis

Kruskal–Wallis test was used, followed either by Dunn’s test, or by the Mann–Whitney test, as appropriate. Results were expressed as mean ± standard error of the mean (SEM). A value of *p* < 0.05 was considered significant. Data analysis was carried out with the GraphPad Prism 6 Program (GraphPad Software, San Diego, CA, USA).

## 5. Conclusions

NGF/TRKA increase the expression of the pro-angiogenic proteins survivin, c-MYC, and VEGF, as well as c-MYC- and β-catenin/TCF-Lef-dependent transcriptional activity by activating the COX-2/PGE_2_ axis in EOC cells. Because EOC cells overexpress NGF/TRKA, which act as proliferative and pro-angiogenic mediators, these results suggest that inhibition of the COX-2/PGE_2_ axis represents an attractive therapeutic approach to prevent angiogenesis and therefore tumor growth in EOC.

## Figures and Tables

**Figure 1 cancers-11-01970-f001:**
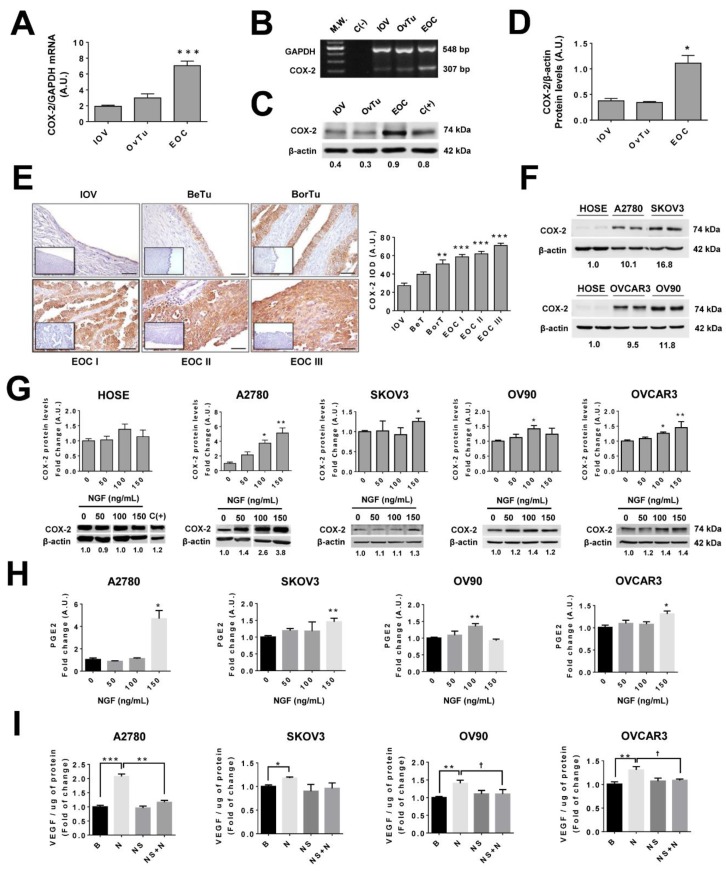
Cyclooxygenase 2 (COX-2) increases during epithelial ovarian cancer (EOC) progression and upon nerve growth factor (NGF) stimulation of EOC cell lines. (**A**) Semi-quantitative analysis of COX-2 mRNA levels in inactive ovarian epithelium (from post-menopausal women, inactive ovarian epithelium (IOV)), ovarian tumors (OvTu) and epithelial ovarian cancers (EOC). *n* = 3, 15, and 10 respectively. *** = *p* < 0.001 with respect to IOV. (**B**) Representative image of agarose gel showing COX-2 products in ovarian samples. M.W: molecular weight. C(−): negative control. (**C**) Representative western-blot of COX-2 protein levels in ovarian tissues (with the respective COX-2/β-actin ratios). (**D**) Quantification of COX-2 protein levels in ovarian biopsies evaluated by western blotting. *n* = 4, 9, and 8 for IOV, OvTu, and EOC, respectively. * = *p* < 0.05 with respect to IOV. (**E**) Immunohistochemical analysis of COX-2 in IOV, OvTu sub-classified into benign tumor (BeT) and borderline tumor (BorT). EOCs were sub-classified into well differentiated epithelial ovarian cancer (EOC I), moderately differentiated epithelial ovarian cancer (EOC II), and poorly differentiated epithelial ovarian cancer (EOC III). Images were obtained at 400× magnification. Negative control: lower left corner. Scale bar: 50 μm. Right: Quantitative analysis of COX-2 immunostaining in ovarian tissues. *n* = 4 for IOV and *n* = 6 or more for the other groups. ** = *p* < 0.01 and *** = *p* < 0.001 with respect to IOV. (**F**) Basal COX-2 immunodetection in ovarian cell lines HOSE, A2780, SKOV3, OV90, and OVCAR3 by western blotting (normalized to the mean COX-2/β-actin ratio). (**G**) COX-2 protein levels after NGF stimulation (50, 100, and 150 ng/mL) for 2 h in HOSE and A2780 cells or 8h in SKOV3, OV90, and OVCAR3 cells (with the COX-2/β-actin ratios). C(+): positive control described in the methodology section. *n* = 4 or more for each condition. * = *p* < 0.05, ** = *p* < 0.01 (**H**) Prostaglandin E2 in culture supernatants of ovarian cell lines after NGF stimulation. *n* = 4 or 5 in duplicate. * = *p* < 0.05 (**I**) Vascular endothelial growth factor (VEGF) protein levels in culture supernatants of EOC cells treated with NGF or the COX-2 inhibitor NS398 (as described in methodology section). B = basal condition (without stimuli); N = NGF; NS = NS398. *n* = 4 or 6 in duplicate. * = *p* < 0.05, ** = *p* < 0.01 and *** = *p* < 0.001 with respect to baseline condition or as indicated (Kruskal–Wallis test and Dunn’s post-test). † *p* < 0.05 with respect to baseline condition or as indicated (Mann–Whitney test). Results are expressed as the mean ± standard error of the mean (SEM).

**Figure 2 cancers-11-01970-f002:**
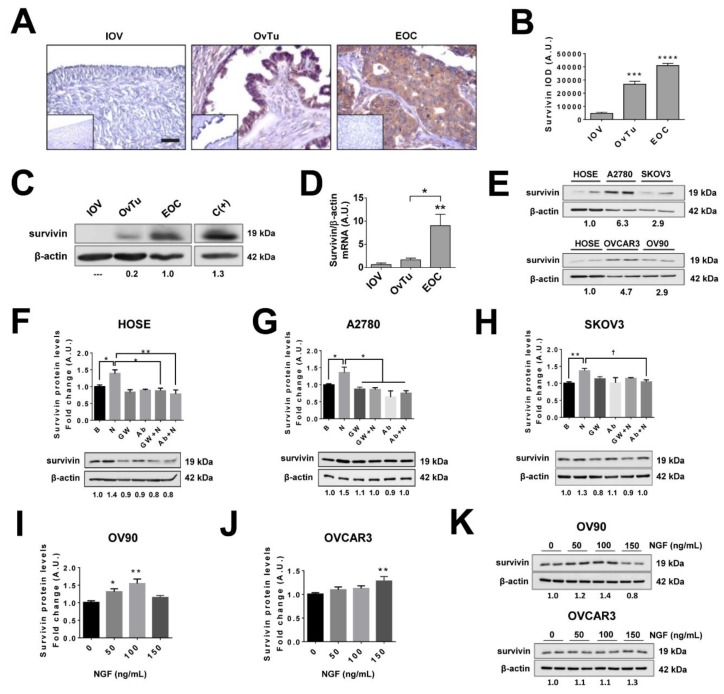
Survivin levels increase during EOC progression and following NGF stimulation of EOC cells. (**A**) Representative images of immunohistochemical detection of survivin in inactive ovarian epithelium (OVI, *n* = 5), epithelial ovarian tumors (OvTu; *n* = 6), and epithelial ovarian cancer (EOC, *n* = 12) tissue. Scale bar: 50 µm. (**B**) Semi-quantification of survivin immunodetection by immunohistochemistry. (**C**) Representative western blot of survivin protein levels in ovarian biopsies (with the survivin/β-actin ratios). (**D**) Quantification of survivin levels detected in extracts of ovarian tissues by western blotting, *n* = 4 for IOV, *n* = 11 for OvTu, and *n* = 15 for EOC. (**E**) Representative images of baseline levels of survivin evaluated by western blotting in ovarian cell lines (normalized to mean survivin/β-actin ratio). (**F**–**H**) survivin protein levels in ovarian cells lines either stimulated with NGF (N; 100 or 150 ng/mL as described in the methodology section), or treated with the neutralizing antibody against-NGF (Ab, 5 µg/mL) or the pharmacological TRKA inhibitor GW441756 (GW, 20 nM) for 2 h (HOSE and SKOV3 cells), 8 h (OV90 and OVCAR3), or 24 h (A2780 cells), respectively (with the survivin/β-actin ratio). *n* = 4 or more in duplicate per condition. (**I**–**K**) survivin protein levels following NGF stimulation in OV90 and OVCAR3 cells (normalized to average survivin/β-actin ratio). *n* = 4 or more in duplicate per condition. * = *p* < 0.05; ** = *p* < 0.01; *** = *p* < 0.001, and **** = *p* < 0.0001 with respect to baseline condition or as indicated (Kruskal–Wallis test and Dunn’s post-test). † = *p* < 0.05 as indicated (Mann–Whitney test). Results are expressed as the mean ± standard error of the mean (SEM).

**Figure 3 cancers-11-01970-f003:**
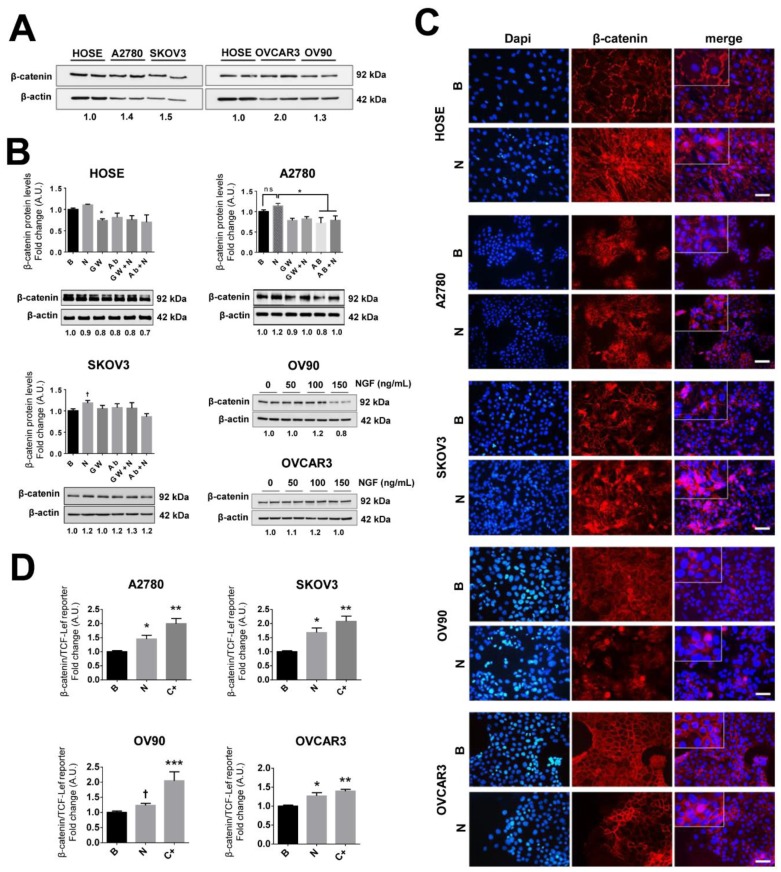
NGF increases β-catenin/ T-cell factor/lymphoid enhancer-binding factor (TCF-Lef) transcriptional activity in EOC cells. Ovarian cell lines were stimulated with NGF (N, 100 or 150 ng/mL, as described), the neutralizing antibody against-NGF (Ab, 5 µg/mL), or the pharmacological TRKA inhibitor GW441756 (GW, 20 nM) for 2 h (HOSE, A2780, and SKOV3 cells) and 8 h (OV90 and OVCAR3). (**A**) Western blots showing baseline levels of survivin in ovarian cell lines (normalized to mean β-catenin/β-actin ratio). (**B**) β-catenin protein levels in extracts of HOSE, A2780, SKOV3, OV90, and OVCAR3 cells analyzed by western blotting (with the β-catenin/β-actin ratios). *n* = 4 or more in duplicate. (**C**) Immunofluorescence of β-catenin in ovarian cell lines either stimulated with NGF (N) or without stimulation (B, basal) to assess the cellular localization of β-catenin. Dapi: 4′,6-diamidino-2-phenylindole (nuclear fluorescent stain). Magnification: 400×. Bar = 50 um. (**D**) β-catenin/TCF-Lef reporter analysis of extracts from EOC cells stimulated with NGF (*n*, 100 or 150 ng/mL) for 24 h. C+: positive control (LiCl 15 mM for 24 h). *n* = 4 or more. * *p* < 0.05, ** *p* < 0.01, and *** *p* < 0.001, with respect to the baseline condition or as indicated (Kruskal–Wallis test and Dunn´s post-test). † *p* < 0.05, with respect to the baseline condition or as indicated (Mann–Whitney test). Results are expressed as the mean ± standard error of the mean (SEM).

**Figure 4 cancers-11-01970-f004:**
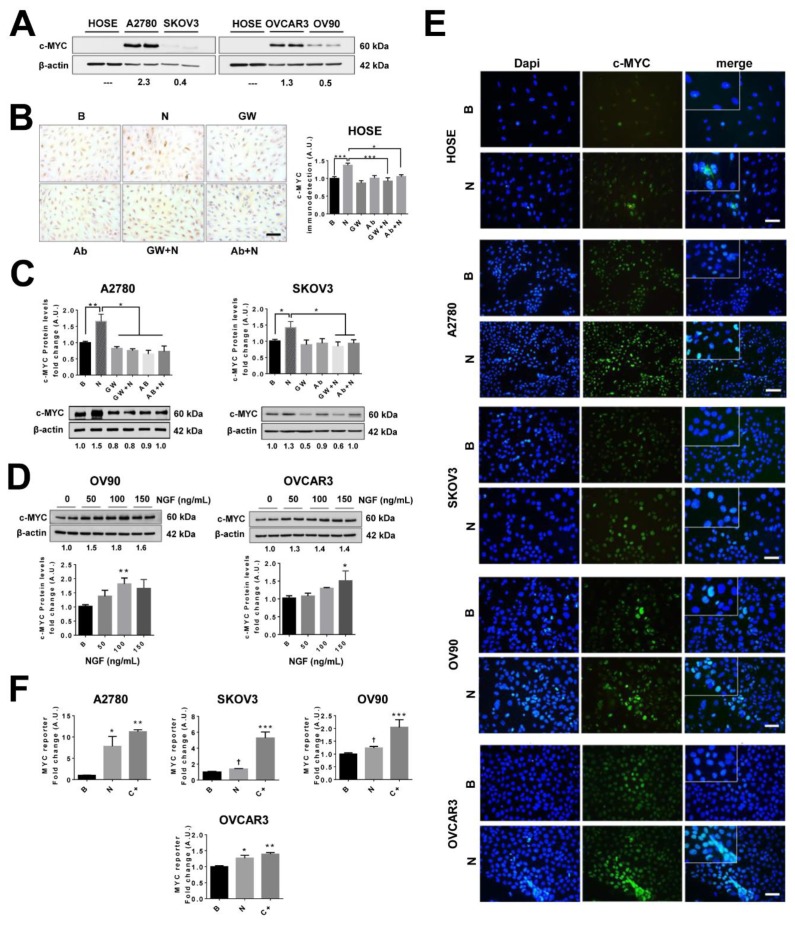
NGF increases c-MYC protein levels and transcriptional activity in EOC cells. Ovarian cells were stimulated with NGF (N, 100 or 150 ng/mL), the neutralizing antibody against-NGF (Ab, 5 µg/mL), or the pharmacological TRKA inhibitor GW441756 (GW, 20 nM) for 2 h (HOSE. A2780 and SKOV3 cells) and 8 h (OV90 and OVCAR3 cells). (**A**) Representative western blot showing baseline levels of c-MYC in ovarian cell lines (compared with average c-MYC/β-actin ratio). (**B**) Immunodetection of c-MYC by immunocytochemistry and semi-quantification in HOSE cells. Bar: 100 μm. *n* = 4; eight pictures per condition were analyzed. (**C**,**D**) c-MYC protein levels assessed by western blotting and quantification (with the β-catenin/β-actin ratio), *n* = 4 in duplicate. (**E**) c-MYC immunodetection in ovarian cells either stimulated with NGF (*n*) or without stimulation (B, basal) to assess the abundance and cellular localization. Magnification: 400×. Bar = 50 μm. (**F**) MYC reporter assay of extracts from EOC cells stimulated with NGF (*n*) for 24 h. C+: positive control (cell co-transfected with reporter construct and c-MYC plasmid). *n* = 4 or more. * = *p* < 0.05, ** = *p* < 0.01, and *** = *p* < 0.001, with respect to baseline condition or as indicated (Kruskal–Wallis test and Dunn´s post-test). † = *p* < 0.05 with respect to baseline condition (Mann–Whitney test). Results are expressed as the mean ± standard error of the mean (SEM).

**Figure 5 cancers-11-01970-f005:**
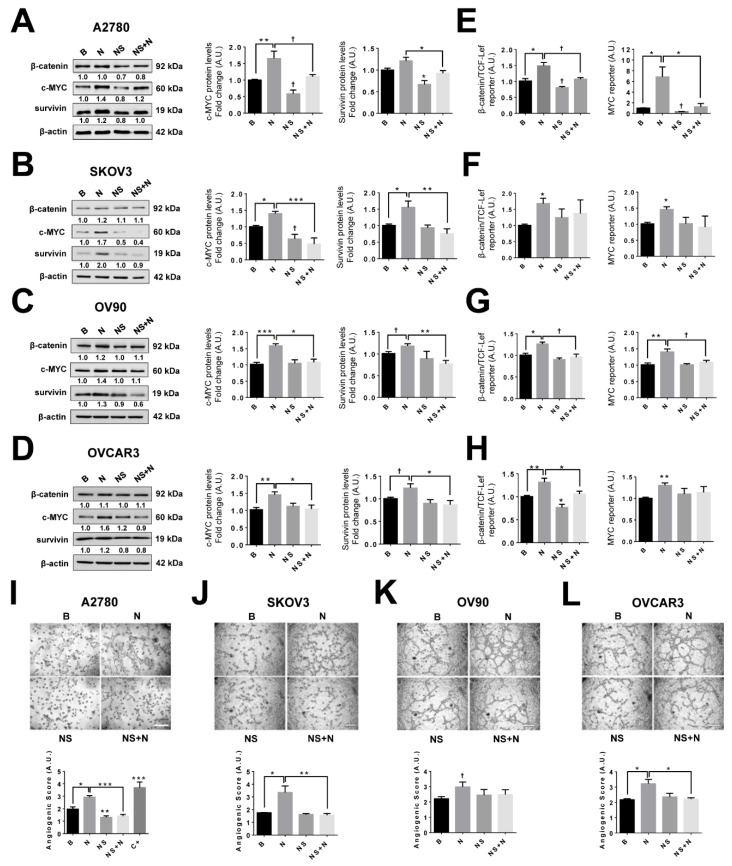
Increased expression of pro-angiogenic proteins stimulated by NGF depends on COX-2/PGE_2_ signaling. EOC cells were treated with the COX-2 inhibitor (NS, 20 μM) for 24 h and then stimulated with NGF (100 or 150 ng/mL) during the last 2 h (HOSE, A2780, and SKOV3 cells) or 8 h (OV90 and OVCAR3 cells). (**A**–**D**) β-catenin, c-MYC, and survivin protein levels evaluated by western blotting and a representative image of results (left) with the respective protein/β-actin ratios. *n* = 4 or more (duplicate). (**E**–**H**) β-catenin/TCF-Lef and MYC transcriptional activity in EOC cells stimulated with NGF either in the absence or presence of the COX-2 inhibitor. (**I**–**L**) Representative images and quantification of the angiogenic score of endothelial cells (EA.hy926) stimulated with conditioned medium from EOC cells treated with the COX-2 inhibitor and NGF. Bar = 100 μm. *n* = 4 (eight pictures per condition). C+: positive control, EAhy926 cells stimulated with VEGF (20 ng/mL, 8 h). B = basal condition (without stimuli); N = NGF 100 or 150 ng/mL, as described in the methodology section; NS = COX-2 inhibitor (NS398, 20 μM). * = *p* < 0.05, ** = *p* < 0.01, and *** = *p* < 0.001, with respect to basal conditions or as indicated (Kruskal–Wallis test and Dunn´s post-test). † = *p* < 0.05, with respect to basal conditions or as indicated (Mann–Whitney test). Results are expressed as the mean ± standard error of the mean (SEM).

**Figure 6 cancers-11-01970-f006:**
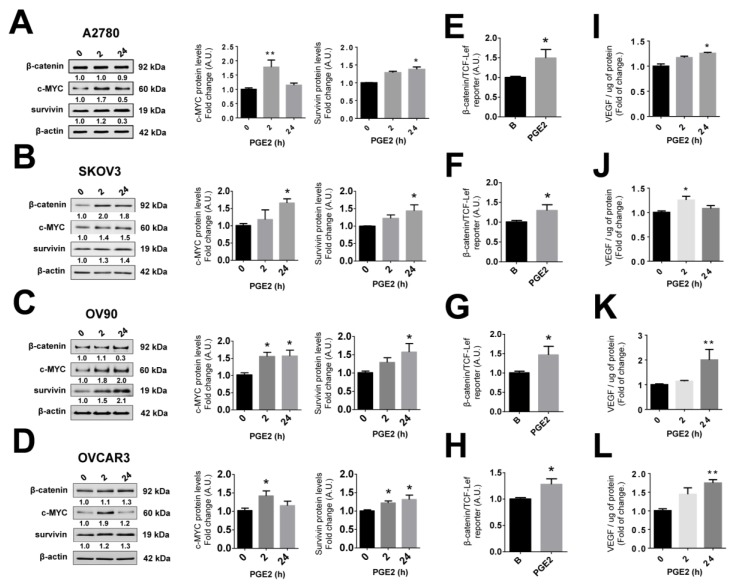
PGE_2_ increases the abundance of pro-angiogenic proteins in EOC cells. EOC cells were treated with prostaglandin E2 (PGE_2_, 20 μM) for 2 or 24 h. (**A**–**D**) β-catenin, c-MYC, and survivin protein levels evaluated by western blotting after PGE_2_ stimulation (with the respective protein/β-actin ratios), *n* = 4 or more (duplicate). (**E**–**H**) β-catenin/TCF-Lef transcriptional activity after PGE_2_ stimulation (24 h), *n* = 4. (**I**–**L**) VEGF protein levels in culture supernatants of EOC cells stimulated with PGE_2_, *n* = 4 in duplicate. * = *p* < 0.05 and ** = *p* < 0.01 with respect to the baseline condition (Kruskal–Wallis test and Dunn´s post-test). Results were expressed as mean ± standard error of the mean (SEM).

**Figure 7 cancers-11-01970-f007:**
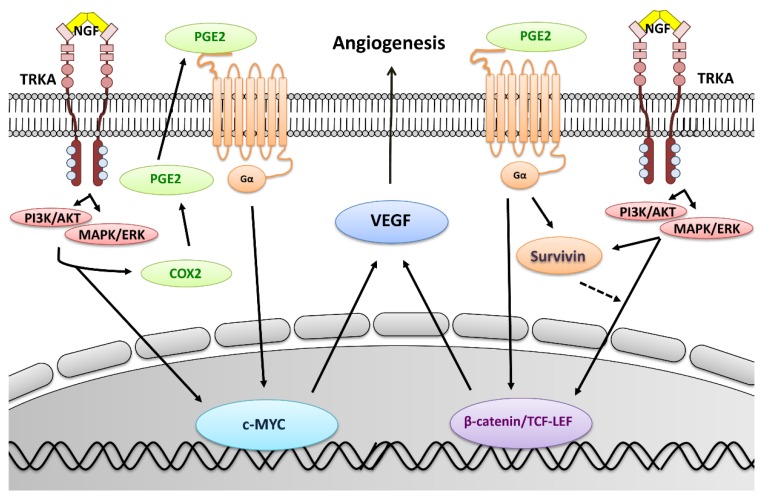
Proposed mechanisms by which NGF increases VEGF expression and angiogenesis in EOC cells. NGF and its high affinity receptor TRKA are overexpressed in EOC cells [9] and activate different signaling pathways, such as phosphoinositide 3-kinase/protein kinase B (PI3K/AKT) and mitogen-activated protein kinase/extracellular signal-regulated kinase (MAPK/ERK) [10,31]. This leads to an increase in COX-2 and prostaglandin E_2_ (PGE_2_), as well as the activation of c-MYC and β-catenin/TCF-Lef-dependent transcription, which increases the expression of VEGF in EOC cells (rapid mechanism). Additionally, autocrine/paracrine signaling via PGE_2_ increases the protein levels and transcriptional activity of c-MYC, as well as the transcriptional activity of β-catenin/TCF-Lef, to augment VEGF expression (delayed mechanism). On the other hand, NGF/TRKA and PGE_2_ increase survivin levels, which stimulates β-catenin/TCF-Lef activity in an amplification loop, as described previously [17], that increases VEGF and therefore the angiogenic potential of EOC cells.

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
