# Peer review of "NGF-Enhanced Vasculogenic Properties of Epithelial Ovarian Cancer Cells Is Reduced by Inhibition of the COX-2/PGE2 Signaling Axis"

_cancers, 2019, doi:10.3390/cancers11121970_

Round 1

Reviewer 1 Report

The revised manuscript has addressed my concerns which I raised in the previous version. The additional data is much better and has strengthened the conclusion of this study.

Author Response

We would like to thank the reviewer for the comments and suggestions. This reviewer was satisfied with the changes we introduced to the original manuscript upon resubmission.

Reviewer 2 Report

The current manuscript by Garrido and co-authors is a revised and resubmitted version which I have reviewed previously. The authors have since then significantly improved their work and addressed the majority of my experiment- and text-related concerns, improved methodology section, figures and figure legends. Most importantly they have strengthened the study by introducing several additional EOC cell lines for the key experiments which showed results in support of the authors’ hypothesis.

Minor comments for the revised version:

I still don’t favor the current title (“vasculogenesis of epithelial ovarian cancer cells”). Wording  “vasculogenesis/angiogenesis OF … cells” implies that the blood vessel initiation/formation occurs from/in these cells. To state so, experiments demonstrating de-differentiation (or other reorganization) of cancer cells into blood vessel-forming cells should be performed.  In the context of current experimental results (increase of transcriptional activity and expression of pro-angiogenic proteins in EOC cells with subsequent mediation of endothelial cell functional behavior by the proteins secreted from EOC cells), the title “angiogenic (or vasculogenic) potential of EOC cells” or “vasculogenic properties of EOC cells” (or similar) is more appropriate. Figures 1-4 are all labeled as figure “N” in the figure legends– please correct. 5I (lower bar graph) – what does C+ (positive control?) stand for here? There is no corresponding image in the A2780 image panel above – please clarify how this control was performed.  Generally, for each figure, the abbreviations must be re-introduced in corresponding legend (e.g. B, N, NS, C+, etc.) regardless their indication in the previous figure legend. Line 423 – please correct grammar.

Author Response

Reviewer 2:

a) I still don’t favor the current title (“vasculogenesis of epithelial ovarian cancer cells”). Wording “vasculogenesis/angiogenesis OF … cells” implies that the blood vessel initiation/formation occurs from/in these cells. To state so, experiments demonstrating de-differentiation (or other reorganization) of cancer cells into blood vessel-forming cells should be performed. In the context of current experimental results (increase of transcriptional activity and expression of pro-angiogenic proteins in EOC cells with subsequent mediation of endothelial cell functional behavior by the proteins secreted from EOC cells), the title “angiogenic (or vasculogenic) potential of EOC cells” or “vasculogenic properties of EOC cells” (or similar) is more appropriate.

Response: We would like to thank the reviewer for the comments and suggestions. We have changed the title of the manuscript to “NGF-Enhanced Vasculogenic Properties of Epithelial Ovarian Cancer cells is Reduced by Inhibition of the COX-2/PGE2 Signaling Axis”.

 b) Figures 1-4 are all labeled as figure “N” in the figure legends– please correct.

Response: The labeling of the figures (Fig.1-4) was corrected as suggested.

 c) 5I (lower bar graph) – what does C+ (positive control?) stand for here? There is no corresponding image in the A2780 image panel above – please clarify how this control was performed.

Response: The information concerning the positive control (EAhy926 cells stimulated with 20 ng/mL of VEGF for 8 h), was added to legend of figure 5.

 d) Generally, for each figure, the abbreviations must be re-introduced in corresponding legend (e.g. B, N, NS, C+, etc.) regardless their indication in the previous figure legend.

Response: To improve comprehension, the abbreviations of cell treatments were re-introduced in the legend of figure 5: “B= basal condition (without stimuli); N= NGF 100 or 150 ng/mL (as described in the methodology section); NS= COX-2 inhibitor (NS398, 20 μM)”

e) Line 423 –please correct grammar.

Response: The manuscript has been revised again for possible mistakes.

Reviewer 3 Report

It is suitable for publication.

Author Response

We would like to thank the reviewer for the comments and suggestions. This reviewer was satisfied with the changes we introduced to the original manuscript upon resubmission.

This manuscript is a resubmission of an earlier submission. The following is a list of the peer review reports and author responses from that submission.

Round 1

Comments from the Editor

  1. While the findings in this study are interesting, the authors failed to use multiple serous ovarian cancer cell lines to verify their findings.

Answer: To improve our work, we replicated the key experiments in another 3 different ovarian cancer cell lines: SKOV3, OV90 and NIH-OVCAR3. According to Domcke et. Al (Nat Commun. 2013;4:2126), OV90 and NIH-OVCAR3 cells present similar characteristics to high grade epithelial ovarian cancer tissue. Please note that these results were included in the text and figures of the manuscript. In addition, we highlighted changes incorporated to address comments by the reviewers.

  1. Furthermore, in addition to HOSE, the authors should consider using fallopian tube epithelial cells as another normal control for their experiments since the fallopian tube is considered as another primary site for ovarian cancer development

Answer: We absolutely agree with the observation, it would be ideal to have a fallopian tube cell line as a control. However, after reviewing the main cell line collections, a non-tumoral cell line with these characteristics is not available. On the other hand, HOSE cells have been used and characterized by several authors, concluding that HOSE cells are useful as control cells for ovarian cancer research (Tsao et al. 1995, Exp Cell Res; Urzua et al. 2012, Horm Metab Res; Shin et al. 2018, PLoS One).

The answers to the comments of the individual reviewers are detailed below.

Reviewer 1 Report

“In Figure 5F, PGE2 increased c-MYC expression after 2 h but not 24 h in A2780 cells. On the other hand, NGF increased in c-MYC protein after 2 h and 24 h. Are there any different mechanism with this finding?”

Answer: We would like to thank the reviewer for the comment. In fact, our results showed that NGF produces quick response in ovarian cells, in which c-MYC levels increase 2 h-post NGF stimulation, which is accompanied by increased PGE2 secretion. PGE2 needs some time to bind its receptor in ovarian cells and increase the c-MYC protein levels. We think that there are two parallel and complementary responses in our in-vitro models, a short-term response (probably mediated by the fast activation of PI3K/AKT and MAPK/ERK pathways by NGF/TRKA in ovarian cells) and a slower response, mediated by PGE2 action. PI3K/AKT and MAPK/ERK modulate the phosphorylation of different amino acid residues in c-MYC, regulating its half-life after short periods (Sears et. Al 2004. Cell Cycle). Because NGF/TRKA activates AKT and ERK signaling pathways in ovarian cancer cells, the final contribution to the increase in c-MYC observed in the present work could be attributable to both mechanisms: the rapid activation of NGF/TRKA signaling pathways and the slightly more delayed NGF-induced PGE2 contribution.

Reviewer 2 Report

This study is interesting for showing the inhibition of COX-2/PGE2 signaling cascade could reduce NGF-mediated angiogenesis in ovarian cancer. However, the whole study was restricted to one HOSE and one ovarian cancer cell model. The A2780 is also not a major serous subtype OvCa cell line. Other major concerns are as follows:

1) The changes in survivin expression in HOSE and A2780 are very hard to see. It should include duplicated lanes for each sample to increase its technical reliability.

Answer: we would like to thank the reviewer for the constructive comments.

We performed experiments in another 3 cell lines to extend our conclusions to other in-vitro models of epithelial ovarian cancer, including high grade serous ovarian cancer cell lines (OV90 and NIH-OVCAR3). Please note that these results were included in the figures of our manuscript.

Regarding the western-blots of survivin, the technique was performed in duplicate; however, for practical proposes the images selected as representative blots show samples in individual lanes. To address this point, we added some figures of blots with duplicate lanes, especially when comparing basal levels of survivin (and the other proteins) in the manuscript. Please, check the new figures added.

2) Besides, why did the western blotting show increased survivin levels in both cell lines at 24 h compared with 2 h in Fig. 3F & 3G?

As described for the reviewer, HOSE cells showed a substantial increase in survivin after brief periods of time (2 h post-NGF stimulation), while for A2780 cells, the greatest increases in survivin levels were observed 24 h after NGF stimulation. This difference could be explained by the abundance of TRKA receptors and the differential signaling pathways promoted in each cell type. As was described for our group, NGF and TRKA increase during EOC progression (Tapia et. al, 2011. Gynecol Oncol), so A2780 has more TRKA receptors and probably a different response pattern than HOSE cells. Besides, it is important to note that EOC cells express more survivin than non-tumoral HOSE cells (Fig. 2E).

As added in the discussion section -to clarify this point- survivin is a short-lived protein with a half-life of about 30 minutes and proteasome inhibitors greatly stabilize survivin (Zhao et al. 2000. J Cell Sci). In addition, survivin levels increase following EGF stimulation via ERK-dependent rapid pathways (2 and 4 h) in beta pancreatic cells (Wang et al. 2010. BMC Mol Biol). These observations suggest that the rapid increase in survivin may be the result of NGF-mediated inhibition of the proteasome in ovarian cells. On the other hand, other authors have shown that ERK and AKT signaling are required to induce survivin transcription in colorectal cancer cells (Ye et al. 2014. Oncogene). Thus, the long-term increase in survivin levels induced by NGF/TRKA in A2780 and the other EOC cells may be attributable to an increase in survivin transcription, which was evidenced by an increase in survivin mRNA post-NGF stimulation in ovarian cells (see Supplementary Fig. 12).

3) Similarly, the changes of b-catenin in Fig. 4A and 4B are too slight to see any difference. They should include qPCR for measuring the gene expression levels when examining the transcriptional activity.

In fact, our findings show that NGF does not change β-catenin levels when compared with the baseline condition (under the experimental conditions tested here). This observation was replicated in most cell lines used. However, the cellular localization of β-catenin is a key point: β-catenin translocation from the plasma membrane to the cytoplasm and then on to the nucleus is important for its function as a transcriptional co-factor. Our results show that although NGF did not change the total β-catenin content, it did change its cellular location (please see the fluorescence images in Fig. 3 that were added to the manuscript), increasing its cytoplasmic / perinuclear location post-NGF stimulation. In accordance with these results, NGF increased β-catenin/TCF-Lef transcriptional activity in all ovarian cancer cell lines used.

Additionally, we performed a qPCR analysis to assess mRNA levels of c-MYC and β-catenin in A2780 cells with and without NGF stimulation. Results are shown below:

As expected, NGF stimulation did not change the mRNA levels of β-catenin, in contrast to c-MYC mRNA, which did increase after NGF stimulation. These results confirm our observations by measuring protein levels.

4) The immunohistochemical analysis and figure quality in Fig. 4C and 4G are too poor to observe any changes in subcellular localization. An amplified picture is expected.

We took new photographs at higher magnification (400X/1000X) and the percentage of positive nuclei were quantified in basal and NGF-stimulated conditions, as shown below. In addition, in the manuscript we show immunofluorescence images obtained using all cell lines. These were added to Fig. 3 (for β-catenin) and Fig. 4 (for c-MYC) of the current manuscript.

Immunocytochemistry of c-MYC and β-catenin in ovarian cells. Percentage of cells with c-MYC positive nuclei (A) and percentage of cells with β-catenin positive nuclei (B) in basal conditions (B, without stimuli) and after NGF stimulation (N, 100 ng/mL, 2h). Bar=50 um. N=4 experiments. *=p<0.05, **=p<0.01 and ***=p<0.001 (Mann Whitney test). ns= non-significant. Results were expressed as mean ± standard error of the mean (SEM).

Reviewer 3 Report

1) All experiments are performed using only one EOC cell line A2780 (endometrioid), not representative of a high grade serous ovarian carcinoma (HGSOC) which is the most common and deadliest type of EOC. Therefore, it is not understood how relevant these data are for the majority of EOC cases. The findings might be simply specific to this cell line only. Alternatively, given similar result trend in the experiments with normal epithelial cells (HOSE), results might be not specific to EOC at all. Expansion to more EOC cell lines will greatly strengthen the study. Expansion with HGSOC cell lines is preferable unless authors want to only focus on endometrioid ovarian adenocarcinoma and clarify that.

Answer: first of all, we would like to thank the reviewer for their comments. We understand the concern and we agree with this point. So, the key experiments were replicated in another 3 different ovarian cancer cell lines: SKOV3, OV90 and NIH-OVCAR3, the latter two being more representative of high grade serous ovarian carcinoma (Domcke et. al. Nat Commun, 2013). Please note that these results were included in the figures of our manuscript, and the modifications suggested were highlighted.

2) For the IHQ of patient samples, designated EOC I-III: is it known what histotypes of ovarian cancer those are? Are they also endometrioid, or mucinous/HGSOG/low-grade/clear cell? This is relevant to help reinforce the in vitro cell line data (Figure 1, 3)

Answer: the samples shown here were from serous epithelial ovarian cancer (low and high grade) patients. Preparations were evaluated by an expert pathologist.

3) Figure 1 and on: statistical representation in the graphs is very hard to follow and makes reading challenging. Authors must unify the designated statistical differences and p-values (only use asterisks or “a,b,c”), correct/unify the labels on the y-axis (e.g. “UA” vs “AU” vs “A.U.” and identify such abbreviations). This comment applies to all figures.

Answer: We performed the respective corrections in the figures.

4) Figure 1: the order of panels and legends do not match, Panel A is not immunohistochemical analysis (which is in panel E) and so on.

Answer: We reviewed this section and the corrections were introduced.

5) How was semi-quantitative analysis of protein levels performed? Assuming that this is a densitometry of Western blot protein bands, this must be described in the M&M section.

Answer: Yes, semi-quantitative analysis of protein levels was performed by scanning densitometry of Western blot bands. We added this information in the methodology section.

6) Upon first introduction of the drugs, the authors should clarify their mode of action and rationality for their dosage. E.g, is 100-150 ng/ml of NGF x 2 hours a commonly used dose and incubation time? Why do some experiments only use 2 hour incubation, while others use both 2 (rapid) and 24 (sustained) hour incubation times? Is 20uM of NS a commonly used dose?

Answer:

To clarify these questions the following:

The doses and times for NGF stimulation were chosen according to previous results from our laboratory indicating that the response of EOC cells to NGF stimulation is very fast. For instance, in ovarian cancer explants, short NGF stimulation (2 h) enhanced VEGF expression and increased the secretion of VEGF from ovarian cancer tissues (Campos et al. 2007, Gynecol Oncol). In addition, other proteins have been shown to be modulated after extended NGF stimulation. For example, NGF/TRKA increases the expression of the FSH receptor after 18 hours of stimulation in human granulosa cells (Salas et al. 2006. J Clin Endocrinol Metab).

Regarding NS398, the dose chosen was previously used by other authors working with different ovarian cancer cells (Urick et al. 2008, Gynecol Oncol. Uddin et al. 2010, Int J Cancer). In addition, the dose of NS398 to be tested in A2780 cells was assessed by measuring inhibition of PGE2 accumulation (see Supplementary Fig. 3). We found 20 uM to represent an appropriate concentration in our model.

7) Figure 2: in panel 2E survivin is barely present in HOSE, while further in panel 2F it is obviously expressed to the extent that it can be further repressed by antibody and inhibitor. This discrepancy must be clarified.

Answer: as observed by the reviewer, HOSE cells express lower baseline levels of survivin than A2780 cells. Thus, to visualize better survivin levels in HOSE cells, we performed western-blotting assays with this cell line using a higher concentration of primary antibody (1:1000 for HOSE cells, 1:3000 for A2780 cells), as described in the methodology section.

8) Figure 3F-G: it appears that antibody and inhibitor repress survivin levels only in the samples that were treated with NGF and only to the basal level of survivin in a particular cell line. There is no additional inhibition of survivin that is endogenously present. This raises a concern that high level of expression of survivin in A2780 cells is totally not dependent on NGF-meditated pathway(s). Also, antibody/inhibitor repress NGF-induced survivin non-selectively, in both non-cancerous and cancerous cells, which is not appropriate if considered for further therapeutic applications. Also, in panel 2F, induction of survivin by NGF is higher at 2hrs than 24hr – the authors could clarify this further in the discussion.

Answer: As indicated by the reviewer, our results did not show that inhibition of NGF/TRKA decreased endogenous survivin levels in ovarian cells. A possible explanation may be that survivin is important because it promotes survival of cancer cells by inhibiting cell death and favoring cell-cycle progression (Reviewed by Mita et al. 2008, Clin Cancer Res). Survivin is regulated by several growth factors and the respective receptors, such as epidermal growth factor (EGF) and fibroblast growth factor (FGF) in breast cancer cells (Asanuma et al. 2005, Cancer Res; Cosgrave et al. 2006. J Mol Endocrinol). These growth factors/receptors are present in ovarian cancer cells, so it is possible that survivin could be regulated by other growth factors in addition to NGF, especially in a compensatory manner when NGF is inhibited. Another interesting observation is that COX-1 is aberrantly expressed in ovarian tumors and ovarian cancer, especially in high grade serous ovarian cancer cells (Daikoku et. al, 2005, Wilson et. al, 2015. Oncotarget). So, following COX-2 inhibition, it is plausible that levels of pro-angiogenic proteins remain high due to the contribution of COX-1.

Regarding the second part of this question, it is important to note that survivin is a short-lived protein with a half-life of about 30 minutes and proteasome inhibitors prevent survivin degradation (Zhao et al. 2000. J Cell Sci). In addition, survivin levels increase following EGF stimulation of pancreatic beta cells via rapid ERK-depending signaling (2 and 4 hours). Thus, in our models of ovarian cancer cells, the rapid increase in survivin could be mediated by NGF-dependent inhibition of proteasome degradation. On the other hand, other authors have described that ERK and AKT signaling is required to activate survivin transcription in colorectal cancer cells (Ye et al. 2014. Oncogene). So the long-term increases in survivin levels could be explained by an increase in the transcription of the protein. This paragraph was added to manuscript, please see the discussion section.

9) Figure 4C: what are B and N (basal and NGF-treated? This must be clarified in the legend) Are these best representative ICQ images? The difference in the nuclear localization (increase) is not obvious, quantification of the images must be presented in a form of a graph. Was this performed on HOSE as well? It would be helpful to see, whether re-localization occurs in non-cancerous cells as well. Quantification for Figure 4G is also required.

Answer: we added the requested clarifications to figure legend. In addition, we visualized c-MYC and β-catenin distribution by immunofluorescence analysis in non-tumor and ovarian cancer cell lines (please, see Fig. 3 and Fig. 4). Finally, we performed a semi-quantification analysis of positive nuclei in A2780 and HOSE cells, as requested:

Immunocytochemistry of c-MYC and β-catenin in ovarian cells. Percentage of cells with c-MYC positive nuclei (A) and percentage of cells with β-catenin positive nuclei (B) in basal conditions (B, without stimuli) and after NGF stimulation (N, 100 ng/mL, 2h). Bar=50 um. N=4 experiments. *=p<0.05, **=p<0.01 and ***=p<0.001 (Mann Whitney test). ns= non-significant. Results were expressed as mean ± standard error of the mean (SEM).

10) Figure 4E: per legend, the graphs represent quantification of IHQ images, providing a panel of representative images here will be helpful.

Answer: We added the representative ICQ panels in the figure (please, see Fig. 4B)

11) Figure 5G: this figure is the most important, as it shows functional relevance of the changes occurring on the mRNA/protein level. Performance of same experiments with HOSE is necessary as well for comparison between non-cancerous and cancerous cells. However, this experiment alone does not prove that pro-angiogenic effect of NGF here occurs through the COX2/PGE2 loop, the authors may want to employ non-selective COX inhibitors or PGE-inhibitor for comparison/reinforcement, as well as try possible COX2-independent mechanisms.

Answer: in order to reinforce our first conclusions, we performed tube-formation assays with endothelial cells using the culture supernatants of the new cell lines that were included (please, see Fig. 5). As expected, COX-2 inhibition prevented the increase in the angiogenic score induced by NGF in almost all the ovarian cancer cell lines tested. Similar results were obtained in the non-tumoral ovarian cell line HOSE, in which COX-2 inhibition prevented the NGF-induced increase in the angiogenic score, as shown in the figure below. However, as clarified in the Supplementary Fig. 9, the basal vasculogenic potential of HOSE cells was lower compared to ovarian cancer cell lines, which correlates with the observed basal levels of secreted VEGF for the different cell lines shown below:

Regarding the second part of this question, our work focusses on studying the contribution of COX-2/PGE2 to pro-angiogenic effects of NGF in ovarian cells, based on preliminary evidence obtained by our group and others. For instance, NGF has been shown to induce COX-2 expression in several cell models, such as mast cells (Murakami et al, 1997 J. Immunol and Sousa-Valente et al. 2018. Osteoarthritis Cartilage). According to the literature, COX-1 is constitutively expressed, unlike COX-2, which is an inducible enzyme. On the other hand, some authors reported that COX-1 inhibition decreases the proliferative and angiogenic potential of ovarian cancer cells (Daikoku et. al, 2005, Wilson et. al, 2015. Oncotarget). Thus, it is highly probably that using a non-selective COX inhibitor, we can expect to observe greater reduction in the expression of pro-angiogenic proteins as well as in the vasculogenic potential in our in-vitro models, compared with only COX-2 inhibition. In addition, if the COX-1 contribution is greater than the COX-2 contribution (which has been described in a high serous ovarian cancer by Wilson et. al, 2015), the use of non-selective COX inhibitors could make it difficult to identify the COX-2 contribution. Finally, although our work favors the interpretation that the effects of NGF / TRKA are mediated by COX-2 / PGE2 signaling, the possibility that NGF could regulate COX-1 under certain conditions cannot be ruled out. Indeed, there is a report in PC12 cells, indicating that COX-1 behaves as a delayed response gene in PC12 cells differentiated by NGF (Kaplan et al. 1997. J Biol Chem). While conceptually very interesting, exploring this possibility in epithelial ovarian cancer cells is beyond the scope of this manuscript.

Minor comments:

12) Angiogenesis means the process of blood vessel formation/development (which may occur in the tissues, including tumor tissues or in the presence of tumor cells), and the wording “angiogenesis of epithelial ovarian cancer cells” in the manuscript title is odd and technically incorrect.

Answer: we improved the title of our manuscript by changing the word angiogenesis to vasculogenesis.

13) Minor grammatical errors throughout text

We carefully reviewed our manuscript and grammatical errors were corrected.

14) All abbreviations must be introduced at first mention (IOV, TRKA, etc.). Also, abbreviations must stay uniform throughout the whole manuscript to avoid confusion (e.g. IOV vs OVI, Ov.Tumors vs Ovarian T., etc.)

Answer: We now used the same abbreviations throughout the manuscript.

15) M&M section: Source and maintenance of SKOV3 cell line must be described.

Answer: We added the missing information concerning SKOV3 cells and their source as well as information concerning the maintenance of the other cell lines incorporated in our work (OV90 and NIH-OVCAR3).

16) M&M section: source of all antibodies for Western blotting must be stated.

Answer: this information was added to the Methodology section.